# Dermal formulation based on carbopol and Gum Arabic improves skin retention of indomethacin

**Hiroko Otake[1], Ogata Fumihiko[1], Yosuke Nakazawa[2], Manju Misra[3,4], Masanobu Tsubaki[5], Naohito Kawasaki[1], Noriaki Nagai** [1]*

1 Faculty of Pharmacy, Kindai University, Higashi-Osaka, Osaka, Japan, 2 Faculty of Pharmacy, Keio University, Minato-ku, Tokyo, Japan, 3 Department of Pharmaceutics, National Institute of Pharmaceutical Education and Research, Gandhinagar, Gujarat, India, 4 Graduate school of Pharmacy, Gujarat Technological University, Gandhinagar Campus Nr. Government Polytechnic K-6 Circle, Gandhinagar, Gujarat, India, 5 Laboratory of Pharmacotherapy, Kagawa School of Pharmaceutical Sciences, Tokushima Bunri University, Sanuki, Kagawa, Japan

* nagai_n@phar.kindai.ac.jp

## Abstract

Top-down approaches efficiently convert hydrophobic drugs into nanoparticles, and the selection of appropriate additives is critical for successful nanoparticle formulation. Methylcellulose is an additive capable of reducing the drug particle size to less than 200 nm using the wet bead milling method, a breakdown method, and a dermal gel containing indomethacin (IMC) nanocrystal formulated with methylcellulose, which achieved high skin absorption. In this study, we focused on gum arabic (GA) as an alternative additive to methylcellulose and demonstrated whether formulations (carbopol gels) containing IMC nanocrystals with GA for dermal application (IMC-NP@GCgel) enhanced the local and systemic absorption of IMC. The particle size was significantly reduced by bead milling with GA, and the mean particle size of the IMC-NP@GCgel was 40–200 nm. The drug release and skin permeability from IMC-NP@GCgel was higher than those from carbopol gels containing the IMC microcrystals (mean particle size was 15.6 μm, IMC-MP@GCgel). In addition, IMC levels in the skin tissue of rats treated with the IMC-NP@GCgel were higher than those of rats treated with the IMC-MP@GCgel. However, the plasma IMC levels did not differ between the IMC-MP@GCgel- and IMC-NP@GCgel-treated rats. We successfully designed IMC nanocrystals using GA instead of methylcellulose. Moreover, we found that the addition of GA supported the absorption of IMC nanocrystals and enhanced the skin retention of the drug without increasing plasma IMC levels. These results provide useful information for the development of dermal formulations based on nanocrystals.

**Data availability statement:** All relevant data are within the paper and its Supporting Information files.

**Funding:** Prof. Dr. Noriaki Nagai, Grant number: KD2304, Funder name: 2023 Kindai University Research Enhancement Grant, URL: https://www.kindai.ac.jp/ The funders had no role in study design, data collection and analysis, decision to publish, or preparation of the manuscript.

**Competing interests:** The authors have declared that no competing interests exist.

## Introduction

Indomethacin (IMC), a non-steroidal anti-inflammatory drug (NSAID), is widely used owing to its anti-inflammatory, analgesic, and antipyretic properties. Its mechanism of action involves the inhibition of cyclooxygenase-2 (COX-2) enzymes, which are critical for prostaglandin biosynthesis. By blocking these enzymes, IMC effectively reduced inflammation, pain, and fever. Thus, its efficacy in reducing inflammation and pain makes it the preferred choice for treating conditions, such as arthritis and other inflammatory disorders [1,2]. Despite these benefits, their clinical application is associated with certain limitations and side effects, particularly when administered orally. As a weakly acidic and poorly water-soluble compound, its oral bioavailability is often suboptimal. Moreover, oral administration of IMC is associated with gastrointestinal (GI) side effects, such as ulcers, bleeding, and discomfort. Furthermore, IMC has a short half-life (4–5 h), necessitating frequent dosing, which increases the risk of side effects [3]. This has led to the exploration of alternative delivery methods, including dermal and transdermal drug delivery systems (TDDS).

TDDS is a compelling alternative to oral and parenteral administration. It provides several advantages, including improved patient compliance, especially for those unable to tolerate oral dosage forms and those suffering from GI issues. By bypassing the hepatic first-pass metabolism and avoiding degradation in the gastrointestinal tract, TDDS ensure consistent serum drug levels and reduce systemic side effects. Additionally, the large surface area and accessibility of the skin make it a convenient route for sustained drug delivery. However, the efficacy of TDDS is often hampered by the natural barrier of the skin, primarily the stratum corneum (SC), which limits drug penetration [4]. Overcoming this barrier is essential for effective dermal and transdermal absorption. Various enhancement techniques have been developed, including the use of chemical permeation enhancers, physical methods such as iontophoresis, and advanced nanotechnology [4–7].

Nanotechnology has emerged as a transformative approach for dermal and transdermal drug delivery, addressing the challenges posed by SC barriers. Nanoparticles (NPs), including nanocrystals (NCs), lipid nanocarriers, and polymeric NPs, offer unique advantages in improving drug permeation and stability [8–12]. These carriers, with sizes ranging from 40 to 800 nm, adhere to the lipid film of SC, increasing the concentration of drug molecules that penetrate deeper into the skin layers [8–12]. A key benefit of NPs is their ability to maintain the integrity of the skin barrier while enhancing drug absorption. This minimizes the risk of skin irritation or damage that is often associated with other enhancement techniques. NPs penetrate the skin via multiple pathways, including intercellular lipid spaces, transcellular routes, and transappendageal routes, such as hair follicles and sweat glands. Collectively, these mechanisms enhance the drug delivery [12–15]. Previous studies have shown that the application of NPs in IMC formulations substantially improves the ocular, intestinal, dermal, and transdermal absorption [11,16–18]. A reduction in particle size increases surface area, enhancing solubility and cellular uptake. This method not only enhances the therapeutic efficacy of IMC but also reduces the required dosage, thereby minimizing systemic side effects. The selection of the gel-base is important

for designing of dermal formulations incorporating NPs, and various hydrogels have been used as TDDS. Among them, Carbopol is one of the most commonly used hydrogels for the preparation of dermal and transdermal formulations, including solid lipid nanoparticles, nanometals, NCs and nanomicelles [19–22]. We also prepared the dispersions containing IMC NCs using methylcellulose as an additive, and Carbopol gels incorporating the OMC NCs enhanced drug absorption by increasing skin penetration [16,17]. Thus, the design of dermal and TDDS-based NCs using Carbopol gel holds great promise for improving the efficacy and safety of IMC and other drugs for dermal and transdermal applications. In contrast, the NCs were prepared by bead milling, and the selection of suitable additives, such as methylcellulose is essential to prepare drug NPs with particle diameter of 200 nm or less using bead milling, and the search for these additives is extremely important in the field of nanomedicine [23]. Gum arabic (GA) is an anionic polysaccharide composed. The molecular weight is approximately $2.5 \times 10^5$ Da, and have d-galactose (main chain) and L-arabinose, L-rhamnose and d-glucuronic acid (side chains). In both hydrophobic peptides and hydrophilic polysaccharide moieties, the GA show the strong emulsification stability, since the GA chemically bind to hydrophobic peptides. GA is used as a base material for powders, suspension agents for cosmetics and beverages, binding agents for tablets, anticrystalline agents, materials for microcapsules, emulsion stabilizers, glazing agents, and coating agents. We demonstrated whether IMC NCs with a particle diameter of 200 nm or less could be produced by bead milling and GA in this study. Moreover, we investigated whether formulations (Carbopol gels) containing IMC NCs with GA for topical application (IMC-NP@GCgel) enhanced local and systemic absorption of IMC.

## Materials and methods

### Reagents

All chemicals used were of the highest purity. Heparin, isoflurane, propyl p-hydroxybenzoate, l-menthol, 10% ammonia solution, GA, and IMC powder were obtained from Wako Pure Chemical Industries Ltd. (Osaka, Japan), while Carbopol® 934 was purchased from Serva (Heidelberg, Germany). Additionally, we utilized a pentobarbital (Tokyo Chemical Industry Co. Ltd., Tokyo, Japan) and Bio-Rad Protein Assay Kit (Bio-Rad, Hercules, CA, USA).

### Animals

Thirty-five male 7-week-old rats (weighing approximately 200 g) were purchased by the Kiwa Laboratory Animals Co., Ltd. (Wakayama, Japan) and divided into nine groups ($5 \times 9 = 45$) for assessing skin irritation ($5 \times 2 = 10$), in vitro transdermal penetration ($5 \times 2 = 10$), in vivo accumulation ($5 \times 2 = 10$), and penetration ($5 \times 3 = 15$) using gel formulations based on IMC microparticles (MPs) and NPs. The rats were housed under standard conditions [7:00 am–7:00 pm (light period); other (dark period); room temperature: 25 °C] and allowed free access to commercial feed (CE-2, CLEA Japan, Inc., Tokyo, Japan) and water. The rats were also kept for one week in an animal room in the environment described above to adapt to the environment. All animal experiments were conducted in compliance with the guidelines of Kindai University, Japanese Pharmacological Society, Guide for the Care and Use of Laboratory Animals provided by the National Institutes of Health, Animal Research: Reporting of In Vivo Experiments (ARRIVE) guidelines, and American Veterinary Medical Association (AVMA) Euthanasia guidelines from 2020. The Animal Experimental Committee of the Kindai University approved the experimental protocol on April 1, 2022 (approval number: KAPS-2022–017). Euthanasia was performed by injecting pentobarbital (200 mg/kg, i.p.) according to the AVMA guidelines 2020 [24]. The human endpoint criteria were bodyweight decreases by more than 10% of the initial weight, and health decline.

### Gel formulations based on IMC MPs and NPs

Gel formulations based on IMC MPs and NPs were prepared using a previously reported breakdown method [11,23,25]. In the pre-preparation, GA was added to a 2 mL tube (TOMY Seiko Co., Ltd., Tokyo, Japan) containing 2 g of zirconia

beads (2 mm diameter) until the tube was filled to approximately 80% of its capacity. The mixture was dry-milled using a Shake Master Neo (1,500 rpm, Biomedical Science Co., Ltd., Tokyo, Japan) for 3 h. Subsequently, IMC suspensions were prepared as follows. A 0.15 g quantity of IMC powder (purchased product) was combined with the milled-GA powder and adjusted to a total volume of 10 mL with purified water to yield IMC suspensions, which were used as IMC-MP suspensions in this study. Another suspension was prepared by triturating 0.15 g of IMC powder using an agate mortar and pestle for 60 min, combining the powder with milled-GA powder, and adjusting the volume to 10 mL with purified water. To produce suspensions containing IMC NPs (IMC-NP suspensions), the suspensions were added to a 2 mL tube containing 2 g of zirconia beads (0.1 mm diameter), filling the tube to approximately 80% of its capacity, and subjected to milling with the Shake Master Neo (1,500 rpm) for 3 h.

In this study, we prepared 1% IMC gels containing IMC microparticles and NPs (IMC-MP@GCgel and IMC-NP@GCgel) using the following procedure: Carbopol® 934 (45 mg) was added to 80 μL of purified water and allowed to swell at 25 °C for 1 h. Subsequently, the l-menthol and IMC-MPs or IMC-NPs suspensions prepared as described above was added. The mixture was neutralized (gelled) using 5% ammonia solution. l-Menthol was incorporated into the gel formulations to enhance the transdermal absorption. The pH of the gels was adjusted to 7.5 by 5% ammonia solution. The compositions of the IMC-MP@GCgel and IMC-NP@GCgel were as follows: 1% IMC, 0.01% GA, 2% l-menthol, and 3% Carbopol in distilled water (w/w%). Our previous study showed that 3% Carbopol® 934 was suitable to use as the gel base for the drug NPs, since the gel showed a high release of drug NPs, and it provided appropriate viscosity [12]. We determined the concentration of Carbopol® 934 by following methods in previous reports.

## High-performance Liquid Chromatography (HPLC)

The IMC was quantified using a HPLC system, LC-20AT (Shimadzu Corp., Kyoto, Japan) [23]. Chromatographic separation employed a mobile phase consisting of acetonitrile and 0.05 M acetic acid in a 70:30 (v/v) ratio, delivered at a flow rate of 0.25 mL/min. Gel samples were prepared by dilution in methanol, while biological specimens (rat skin) underwent homogenization in methanol followed by centrifugation at $20,400 \times g$ for 20 min at 4 °C. Blood samples were centrifuged under identical conditions. The resulting supernatants were analyzed to determine IMC concentrations in both skin tissue and blood. For each analysis, 100 μL of the sample was transferred into a sample vial, and 50 μL of methanolic propyl p-hydroxybenzoate in methanol (used as an internal standard) was added. A 10 μL aliquot of the mixture was injected *via* an SIL-20 AC auto-sampler. Chromatographic separation occurred at 35 °C using an Inertsil® ODS-3 column housed in a CTO-20 AC oven (GL Science Co., Inc., Tokyo, Japan), with detection performed at a wavelength of 254 nm. The IMC peak was observed at 10.1 min, whereas the internal was eluted at 3.7 min. The detection limit for IMC was set at 120 ng/mL.

## Particle size and distribution of IMC gel formulations

The gels incorporating IMC were diluted 1,000-fold with purified water, and the particle size and distribution were measured using a SALD-7100 (Shimadzu Corp., Kyoto, Japan) and Nanoparticle Tracking Analysis NanoSight LM10 (Quantum Design Japan, Tokyo, Japan). For the SALD-7100 measurements, the scattering light intensity was maintained within a range of 40–60% of the maximum, with a refractive index set to 1.60 ± 0.10i. For the NanoSight measurements, the conditions were set to a wavelength of 405 nm (blue), a measurement duration of 60 sec, and a viscosity of 1.27 mPa·s to evaluate the drug NPs [12,25].

## Crystalline form of IMC in the gel formulations

The IMC-MP@GCgel and IMC-NP@GCgel were diluted 10-fold with purified water, and subjected to lyophilization using a FREEZE DRYER FD-1000 (TOKYO RIKAKIKAI Co., Ltd., Tokyo, Japan). The process conditions were set at a temperature of −20 °C, a pressure of 20 Pa, and a duration of 48 h. The crystalline characteristics of the freeze-dried samples

were analyzed using powder X-ray diffraction (XRD) and differential thermal analysis (TG-DTA). For the XRD analysis, a Mini Flex II system (Rigaku Co., Tokyo, Japan) was employed, with diffraction angles scanned over a range of 5° to 60° at a rate of 10°/min. TG-DTA measurements were conducted using a simultaneous thermal analyzer, DTG-60H (Shimadzu Corp.) with approximately 5 mg of the powdered sample. The analysis was performed under a nitrogen atmosphere with the temperature ramped from 50 °C to 200 °C at a constant heating rate of 10 °C/min [25,26].

## Viscosity of IMC gel formulations

The viscosities of the IMC-MP@GCgel and IMC-NP@GCgel were evaluated using an SV-1A viscometer (A&D Company Limited, Tokyo, Japan). Approximately 0.3 g of each sample was subjected to measurement at a rotational speed of 60 rpm and a temperature of 22 °C for a duration of 3.5 min [25].

## Drug solubility in IMC gel formulations

The IMC-MP@GCgel and IMC-NP@GCgel were diluted 10-fold with purified water at 22 °C. Subsequently, IMC was fractionated into its soluble and non-soluble forms *via* centrifugation at $1 \times 10^5$ g using a Beckman Optima™ MAX-XP Ultracentrifuge (Beckman Coulter, Osaka, Japan). The IMC content in the resulting supernatants was quantified by HPLC analysis following a previously described method [25].

## Uniformity of IMC gel formulations

A 0.3 g sample of IMC-MP@GCgel and IMC-NP@GCgel was subdivided into 10 equal portions (0.03 g each) and dissolved in methanol. The resulting solutions were filtered using a 450-nm membrane filter, and the IMC content in each filtrate was quantified *via* HPLC, employing a previously described protocol. To assess the uniformity of the gels, the standard deviation (S.D.) of the IMC content across 10 subdivisions was calculated, representing the degree of nonuniformity of the IMC distribution within the gel [25].

## Stability of IMC gel formulations

A 0.3 g sample of IMC-MP@GCgel and IMC-NP@GCgel was subdivided into 10 equal portions (0.03 g each) and stored in a sealed and dark environment for one month. After that, the non-uniformity, solubility, and viscosity were measured using methods described above.

## Skin irritation of IMC gel formulations

Ten Seven-week-old Wistar rats were allocated to two groups (five rats per group), and the abdominal skin of each rat was shaved using an electric clipper and razor one day prior to the experiment. The 0.3 g of IMC gels were evenly applied to a 2 cm² area of the abdominal skin once daily at 10:00 am for a consecutive period of 1 month. After that, the skin irritation on the abdominal skin was evaluated based on redness and skin damage through visual inspection. The evaluation was conducted 24 h after the final gel application. During the experimental period, hair shaving was performed once every five days.

## *In Vitro* IMC release from gel formulations

A DURAPORE membrane filter (pore size: 0.45 µm) was mounted onto a Franz diffusion cell with an effective diffusion area of 2 cm². The receptor (lower) chamber of the cell, with a capacity of 12.2 mL, was filled with 0.2 mM phosphate buffer (prepared by combining 28 mL of 0.2 mM $NaH_2PO_4$ and 72 mL of 0.2 mM $Na_2HPO_4$, and diluting the mixture to 200 mL with purified water; pH 7.2). A 0.3 g sample of either IMC-MP@GCgel or IMC-NP@GCgel was applied to the donor (upper) compartment, which was subsequently sealed with a stopper and wrapped in aluminum foil to minimize

evaporation and contamination. The receptor chamber was maintained at 37 °C and stirred continuously with a magnetic stirrer to ensure homogeneity. At designated time intervals (0.5, 1, 2, 3, 6, and 24 h), 200 µL aliquots were withdrawn from the receptor chamber and replaced with an equal volume of fresh 0.2 mM phosphate buffer. Each collected sample (50 µL) was mixed with 100 µL of methanol, and the IMC concentration was quantified using HPLC described above. The particle size distribution of the formulations was assessed using NanoSight. The area under the IMC release concentration-time curve ($AUC_{0\text{-}24h\ release}$) was computed using the trapezoidal rule up to the final sampling point (24 h) [12].

### *In Vitro* transdermal penetration of IMC gel formulations

Ten Seven-week-old Wistar rats were allocated to two groups (five rats per group), and the abdominal skin of each rat was shaved using an electric clipper and razor one day prior to the experiment. After 24-h period, the rats were euthanized using a lethal dose of pentobarbital (200 mg/kg). The abdominal skin was excised and mounted onto a Franz diffusion cell to evaluate the *in vitro* skin penetration of the IMC gel, following the methods outlined in previous studies [12,25]. The receptor chamber of the diffusion cell (12.2 mL) was filled with 0.2 mM phosphate buffer (pH 7.2), and 0.3 g of IMC gel was uniformly applied to the membrane surface (area: 2 cm²). The system was maintained at 37 °C for 24 h. At designated time points (0.5, 1, 2, 3, 6, and 24 h), 100 µL samples were collected from the receptor chamber. The IMC concentration and particle size distribution in the samples were analyzed using HPLC and NanoSight, respectively, as previously described. The area under the IMC concentration-time curve for skin penetration ($AUC_{0\text{-}24h\ penetration}$) was calculated using the trapezoidal rule up to 24 h. Additionally, pharmacokinetic parameters, including the penetration rate ($J_c$), skin/preparation partition coefficient ($K_m$), skin penetration coefficient ($K_p$), diffusion constant within the skin ($D$), lag time ($\tau$), skin thickness ($\delta$, 0.071 cm), and the amount of IMC in the receptor chamber at time $t$ ($Q_t$), were determined based on equations Eq. 1 and 2 [12,27]:

$$J_c = \frac{Q}{A \cdot (t\text{-}\tau)} = \frac{D \cdot K_m \cdot C_c}{\delta} = K_p \cdot C_c \tag{1}$$

$$D = \frac{\delta^2}{6\tau} \tag{2}$$

### Percutaneous absorption of IMC gel formulations

Fifteen Seven-week-old male Wistar rats were divided into three groups (five rats per group). Two groups were used to measure the dermal absorption of IMC-MP@GCgel and IMC-NP@GCgel, and the other group was used to analyze the pharmacokinetic parameters. During pretreatment, the rats were anesthetized with isoflurane (flow rate: 1.0 L/min, concentration: 3%) and jugular vein cannulation was performed. Under anesthesia, the rats were placed in the supine position, and a 1.5 cm incision was made on the right side of the neck. Muscle layers were carefully separated using tweezers to expose the external jugular vein. A catheter (Faicon tube, SH No. 00) prefilled with heparin solution (10 IU/mL heparin sodium) was passed subcutaneously and externalized through the dorsal neck. The catheter is connected to a syringe for blood sampling. Two ligatures were applied to the external jugular vein: the proximal (cardiac) side was temporarily occluded to halt blood flow, while the distal (cervical) side was tightly secured. A small incision was made in the vein and approximately 3 cm of the catheter was inserted. Proper insertion was confirmed by observing the blood flow into the catheter, after which the proximal ligature was tightened. The incision site is sutured to complete the procedure. After 24 h, 0.3 g of IMC gels were evenly applied to a 2 cm² area of the abdominal skin. Blood samples (200 µL each) were collected from the right jugular vein via the cannulation tube at predetermined time points: 0.5, 1, 2, 3, 6, and 24 h post-application. The collected blood was centrifuged at 20,400 × g for 20 min at 4 °C, and the resulting plasma was analyzed for IMC

concentration using HPLC described above. The area under the plasma IMC concentration-time curve ($AUC_{0-24h\ plasma}$) was calculated using the trapezoidal rule for up to 24 h. The pharmacokinetic parameters were subsequently determined using Eq. 3 and 4 [12,27].

$$C_{IMC} = C_0 \cdot e^{-k_e \cdot t} \tag{3}$$

$$C_{IMC} = \frac{k_a \cdot F \cdot D}{V_d(k_a - k_e)}\left(e^{-k_e \cdot (t-\tau)} - e^{-k_a \cdot (t-\tau)}\right) \tag{4}$$

The pharmacokinetic parameters $k_e$ and volume of distribution ($V_d$) were determined using Eq. 3, based on plasma IMC concentrations following a single intravenous administration of 0.3 mL IMC solution (200 µg/kg) into the femoral vein. The initial plasma concentration ($C_0$), elimination rate constant ($k_e$), and $V_d$ were calculated as 2.68 ± 0.13 µg/mL, 0.05 ± 0.07 h$^{-1}$, and 52.1 ± 1.98 mL/kg, respectively (n = 5). In the percutaneous absorption study, the absorption rate constant ($k_a$) and bioavailability ($F$) were estimated using Eq. 4.

### Accumulation of IMC gel formulations in skin tissue

Ten seven-week-old Wistar rats were assigned to two groups (five rats per group). One group was used to evaluate the IMC-MP@GCgel, and the other group was used to evaluate the IMC-NP@GCgel. One day prior to the experiment, the abdominal skin of each rat was carefully shaved using an electric clipper and razor. After a 24-h acclimation period, 0.3 g of IMC gel was uniformly applied to a 2 cm² area of the abdominal skin. At 6 h post-application, the rats were euthanized by intraperitoneal injection of a lethal dose of pentobarbital (200 mg/kg). The abdominal skin was excised, and any IMC gel remaining on the skin surface was removed. The excised skin was homogenized in methanol to extract IMC. The resulting homogenates were centrifuged at 20,400 × g for 20 min at 4 °C. The IMC concentration in the supernatant was subsequently determined using HPLC as described above. These operations were performed as described in a previous study [27]

### Statistical analysis

Statistical analyses were conducted using the JMP software version 5.1 (SAS Institute). Data are presented as the mean ± standard error of the mean (SEM). Comparisons were performed using two-way repeated-measures analysis of variance (ANOVA) in the results of Figs 4A, 5A, and 6A, and one-way ANOVA followed by post-hoc tests, including the unpaired Student's t-test in other results. A $P$-value of <0.05 was considered statistically significant.

## Results

### Preparation and characterization of IMC-MP@GCgel and IMC-NP@GCgel

Fig 1 shows the particle size distributions of IMC-MP@GCgel and IMC-NP@GCgel was analyzed. Following bead milling, a substantial reduction in particle size was achieved. The mean particle sizes of IMC-MP@GCgel and IMC-NP@GCgel by SALD-7100 were measured as 15.6 ± 0.35 µm (Fig 1A) and 0.089 ± 0.018 µm (Fig 1B), respectively. Additional analysis of IMC-NP@GCgel using the NanoSight revealed a mean particle size of 109.7 ± 3.8 nm (Fig 1C). The images supported the quantitative data of drug uniformity in the gel (Fig 3A), and suggested that the particles were more evenly distributed in the IMC-NP@GCgel than in the IMC-MP@GCgel (Fig 1D and 3A). Fig 2 shows the XRD and TG-DTA patterns used to evaluate the crystal structures and thermal properties of the IMC gels. The XRD peaks were detected at the same positions in both formulations (IMC-MP@GCgel and IMC-NP@GCgel). No changes in the TG patterns were observed in the vehicle, IMC-MP@CGgel, and IMC-NP@CGgel, confirming that no thermal degradation and weight loss patterns occurred

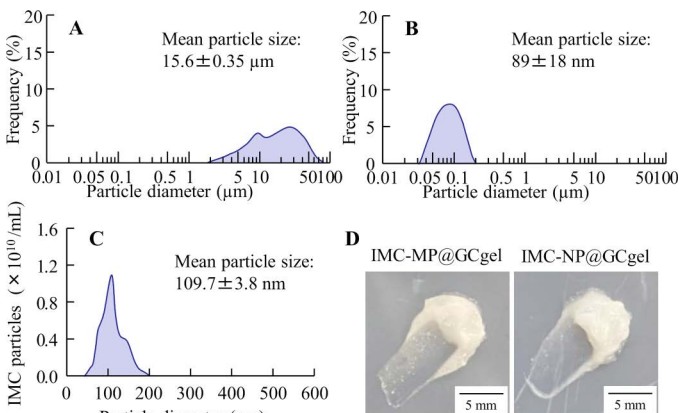

**Fig 1. Particle size frequency and image of IMC-MP@GCgel and IMC-NP@GCgel.** (A) and (B) Particle size distribution data of the IMC-MP@GCgel (A) and IMC-NP@GCgel (B) determined using SALD-7100. (C) Particle size distribution of the IMC-NP@GC gel using NanoSight LM10. (d) Digital images of IMC-MP@GCgel and IMC-NP@GCgel. The particle size was significantly reduced by bead milling with GA, and the mean particle size of the IMC-NP@GCgel was 40–200 nm.

within the measured temperature range. On the other hand, the DTA pattern within the same temperature range detected an endothermic peak at approximately 162 °C, which, in conjunction with the TG results, was identified as the melting point. Furthermore, the melting points of IMC-MP@CGgel and IMC-NP@CGgel were similar. Both the XRD and TG-DTA results indicated no significant changes in the crystal patterns across the raw IMC powder, vehicle, IMC-MP@GCgel, and IMC-NP@GCgel. Fig 3 shows the uniformity, viscosity, and solubility of the IMC gels. The viscosity revealed no significant differences among the vehicle, IMC-MP@GCgel, and IMC-NP@GC gels (Fig 3C). However, the uniformity and solubility of IMC-NP@GCgel were higher than those of IMC-MP@GCgel (Figs 3A and 3B). These results suggest that bead milling effectively reduces the IMC particle size without altering its crystal structure. Moreover, particle size reduction improved uniformity and solubility, whereas viscosity remained unaffected across the formulations. Table 1 show the stability and toxicity of IMC@GCgels. There was no significant change in particle size, uniformity, solubility and viscosity were similar between immediately and one month after preparation. Furthermore, no redness or other visible signs of irritation were observed on the abdominal skin one month after the final application (not detected, N.D.).

The particle size, non-uniformity, solubility, and viscosity were measured 1 month after preparation. The formulations were applied to rats once daily at 10:00 am for a consecutive period of one month to evaluate the skin irritation of IMC@GCgels. n = 5. N.D., Not detected. *$P < 0.05$ vs. IMC-MP@GCgel for each category (Student's t-test).

### Changes in drug release, retention, permeability, and systemic absorption profiles of IMC-MP@GCgel and IMC-NP@GCgel

The drug release profiles of IMC-MP@GCgel and IMC-NP@GCgel are shown in Fig 4. Both formulations exhibited sustained release over 24 h. Drug release was initiated within 0.5 h, with a significantly higher release observed in the IMC-NP@GCgel than in the IMC-MP@GCgel (Fig 4A). The levels of $AUC_{0-24h\ release}$ was also significantly greater for the IMC-NP@GCgel (Fig 4B). In addition, some of the IMC particles released from IMC-NP@GCgel were maintained NPs state, and the mean particles size was 154.7 ± 10.1 nm (Fig 4C and 4D). The *in vitro* transdermal permeability profiles of the IMC gels are presented in Fig 5, and the kinetic parameters are summarized in Table 2. The IMC can pass through the skin, and both formulations displayed linear permeability, with drug permeation detectable as early as 0.5 h. The IMC-NP@GCgel group demonstrated higher drug permeation levels, including $AUC_{0-24h\ penetration}$, $J_c$, $K_p$, and $K_m$, compared to the IMC-MP@GCgel group. On the other hand, the IMC NPs dissolved during the skin permeation process, since the

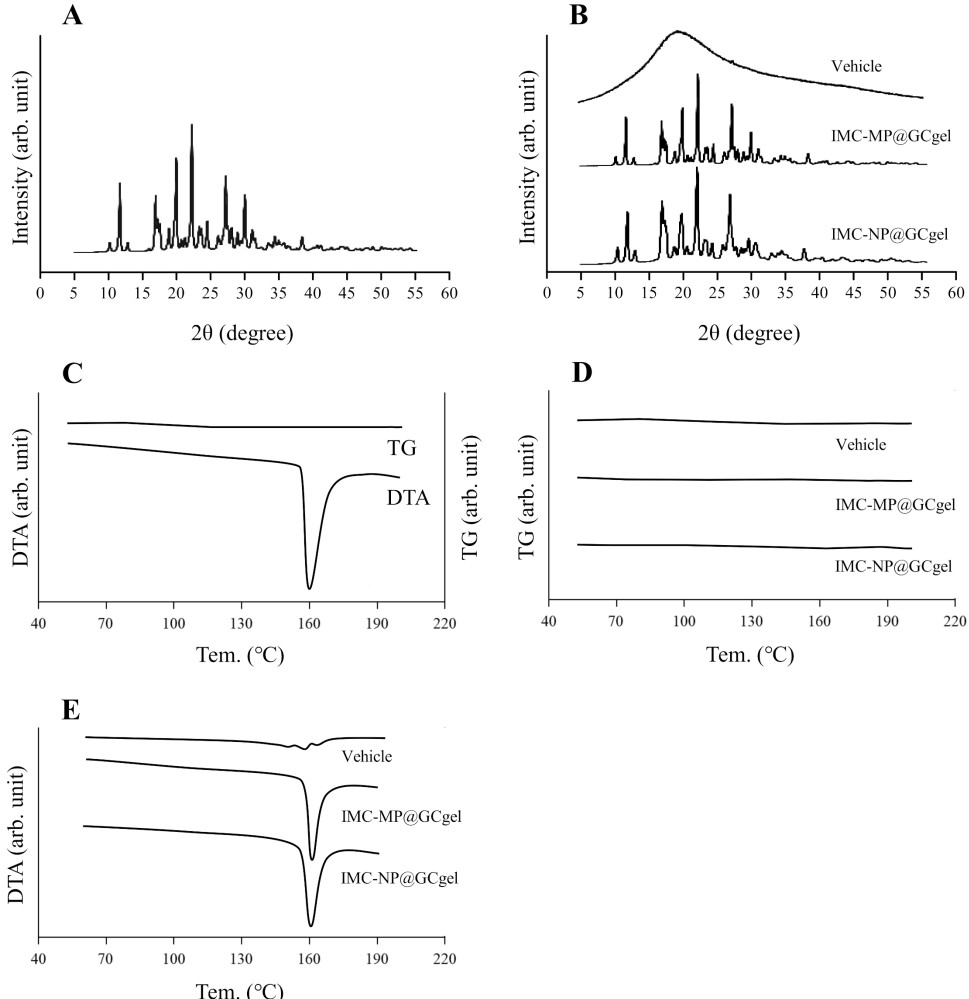

**Fig 2. XRD and TG-DTA patterns of IMC-MP@GCgel and IMC-NP@GCgel.** (A) XRD pattern of IMC powder. (B) Changes in XRD pattern of vehicle, IMC-MP@GCgel and IMC-NP@GCgel. (C) TG and DTA pattern of IMC powder. (D) and (E) Changes in TG (D) and DTA pattern (E) of vehicle, IMC-MP@GCgel and IMC-NP@GCgel. No difference in the XRD and TG-DTA patterns were observed in IMC-MP@GCgel and IMC-NP@GCgel.

IMC NPs were not detected in the reservoir chamber after application. Fig 6A illustrates the plasma concentration profiles following gel application. The pharmacokinetic parameters are summarized in Table 3. No significant differences were observed in the $AUC_{0-24h\ plasma}$, $k_a$ and $F$, between the two formulations (Student's t-test). However, the IMC-NP@GCgel group exhibited significantly higher skin retention than the IMC-MP@GCgel group, suggesting enhanced localized drug delivery (Fig 6B).

## Discussion

In this study, we designed Carbopol gel formulations containing IMC NCs and GA (IMC-NP@GCgel) and demonstrated whether their skin permeability was increased. The results showed that an IMC-NP@GCgel with an average particle size of 109.78 nm was successfully prepared using the breakdown method (wet bead milling) with GA as the additive. In addition, we found that the affinity between skin and drugs in rats treated with IMC-NP@GCgel containing GA was higher than that ($J_c$ 158 nmol/cm²/h, $K_p$ 5.7 × 10⁻⁴ cm/h, $K_m$ 10.8 × 10⁻², $\tau$ 2.3 h, and $D$ 3.7 × 10⁻⁴ cm²/h) of the previously reported

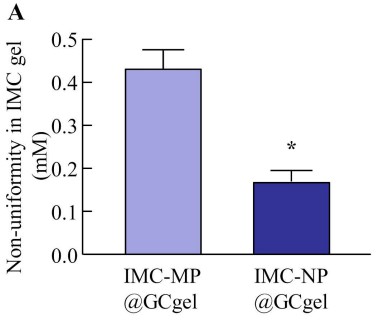
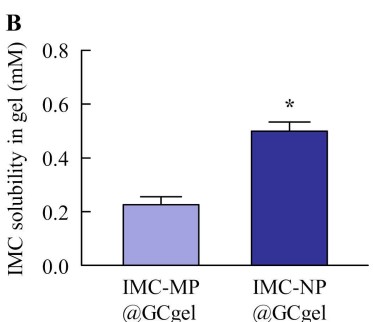

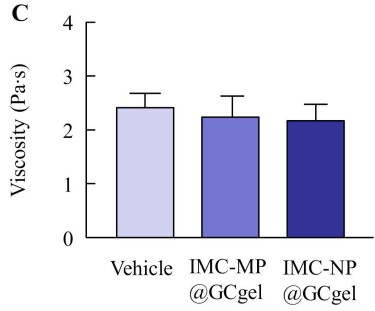

**Fig 3. Non-uniformity (A), solubility (B), and viscosity (C) of IMC-MP@GCgel and IMC-NP@GCgel.** n = 5. *P < 0.05 vs. IMC-MP@GCgel for each category (Student's t-test). The uniformity and solubility of IMC in IMC-NP@GCgel were higher than those in IMC-MP@GCgel. However, the viscosity of IMC was similar in the IMC-MP@GCgel and IMC-MP@GCgel.

**Table 1. Evaluation of formulation safety and stability of IMC-MP@GCgel and IMC-NP@GCgel.**

| Formulation | Mean particle size (μm) | Non-uniformity (mM) | IMC solubility (mM) | Viscosity (Pa·s) | Skin irritation |
|---|---|---|---|---|---|
| IMC-MP@GCgel | 17.2 ± 0.41 | 0.42 ± 0.05 | 0.22 ± 0.05 | 0.23 ± 0.06 | None |
| IMC-NP@GCgel | 0.98 ± 0.02 | 0.18 ± 0.02* | 0.49 ± 0.03* | 0.22 ± 0.05 | None |

carbopol gels containing IMC NCs with methylcellulose (MC) as additives [16,17], suggesting usefulness application potentials for localized drug delivery.

Formulations utilizing NCs have significant potential in nanomedicine and top-down methods are commonly employed for their production. When the bead-milling technique is used to obtain drug NCs smaller than 200 nm, cellulose is typically required as a stabilizing additive. In contrast, the use of alternative additives generally results in NCs in the range 200–400 nm [28–31]. Therefore, identifying additives capable of producing NCs and elucidating their pharmacokinetic properties may advance the development of NC-based nanomedicines. To address this issue, we explored methods to prepare NCs without relying on cellulose.

First, IMC nanoparticle-containing carbopol gels were prepared using a bead-milling method and additives such as GA and l-menthol. l-Menthol was used because it is necessary to promote the skin permeability of NCs [12,32]. IMC NPs were prepared by employing GA as an additive and bead milling for 3 h. Notably the selection of additives is critical for achieving nanosized particles during wet milling. Although, the MC was used to prepare the IMC NPs in the previous our reports [16,17], the use of GA in this study facilitated the successful preparation of IMC NPs with an average particle size of 109.78 nm (Fig 1), demonstrating its utility as an additive for preparation of NPs *via* wet milling. Moreover, evaluation of the crystalline state following milling revealed no changes in the crystal patterns, as determined by XRD and TG-DTA

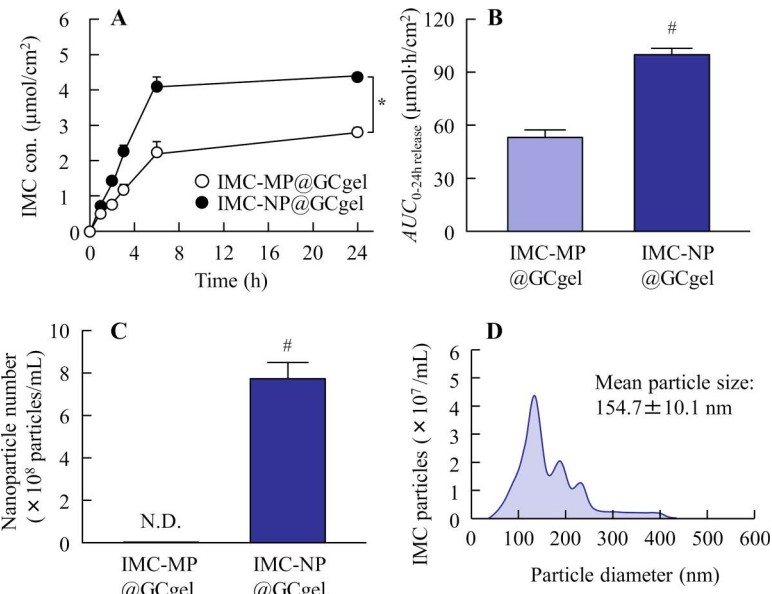

**Fig 4. Release behavior of IMC from IMC-MP@GCgel and IMC-NP@GCgel through a 450-nm pore membrane.** (A) and (B) Drug release (A) and $AUC_{0-24h\ release}$ (B) from the IMC-MP@GC gel and IMC-NP@GC gel. (C) and (D) Number (C) and size distribution (D) of IMC NPs in the reservoir chamber 24 h after the application of IMC-NP@GCgel. n = 5. N.D., not detectable. *$P < 0.05$ vs. IMC-MP@GCgel (repeated-measures ANOVA). #$P < 0.05$ vs. IMC-MP@GCgel for each category (Student's t-test). Drug release from the IMC-NP@GC gel was higher than that from the IMC-MP@GCgel, and IMC NPs (40–430 nm) were detected in the reservoir chamber after application of the IMC-NP@GCgel.

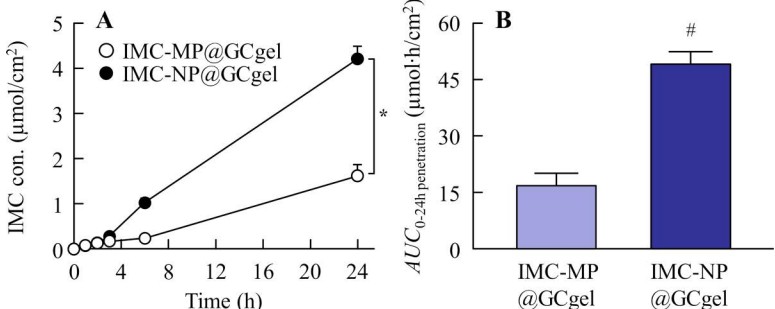

**Fig 5. Changes in in vitro transdermal penetration of IMC from IMC-MP@GCgel and IMC-NP@GCgel through a rat skin.** (A) and (B) Trans-dermal penetration (A) and $AUC_{0-24\ release}$ (B) of IMC-MP@GCgel and IMC-NP@GCgel (in vitro study). n = 5. *$P < 0.05$ vs. IMC-MP@GCgel (repeated-measures ANOVA). #$P < 0.05$ vs. IMC-MP@GCgel for each category (Student's t-test). Drug penetration from the IMC-NP@GCgel was higher than that from the IMC-MP@GCgel. In contrast to the result obtained for the release of IMC from the IMC-NP@GCgel, IMC NPs were not detected in the reservoir chamber after application.

(Fig 2). As the XRD and TG-DTA patterns remained consistent, no changes in the crystal structure were observed after gel production. The drug solubility was also assessed. Consistent with the predictions based on the Ostwald-Freundlich equation [33–35], nanosizing led to an increase in solubility (Fig 3B). Additionally, the viscosities of the IMC-MP@GCgel and IMC-NP@GCgel were comparable (Fig 3C), and the uniformity of IMC in the IMC-NP@GCgel was significantly higher than that in the IMC-MP@GC gel (Fig 3A). Furthermore, no changes were observed in the characterization, such as drug particle size, uniformity, solubility, and viscosity over the one-month period following gel preparation, and skin toxicity was not observed through visual inspection after repetitive application (Table 1). From these results, it was suggested that

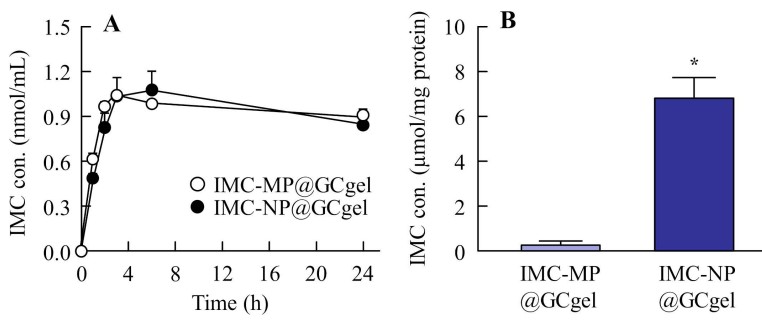

**Fig 6. Changes in IMC levels in blood and skin tissue of rat treated with IMC-MP@GCgel and IMC-NP@GCgel.** (A) Plasma IMC levels in rats 24 h after treatment with IMC-MP@GCgel and IMC-NP@GCgel. (B) IMC content in the skin tissue of rats 6 h after treatment with IMC-MP@GCgel and IMC-NP@GCgel. n = 5. *$P < 0.05$ vs. IMC-MP@GCgel for each category (student's t-test). IMC levels in the skin tissue of rats treated with the IMC-NP@GCgel were higher than those treated with the IMC-MP@GCgel. However, the plasma IMC levels were not difference between IMC-MP@GCgel- and IMC-NP@GCgel-treated rats.

**Table 2. Pharmacokinetic analysis of IMC@GCgel in *in vitro* skin penetration.**

| Formulation | $J_c$ (μmol/cm²/h) | $K_p$ (×10⁻³ cm/h) | $K_m$ (×10⁻¹) | $\tau$ (h) | $D$ (×10⁻³ cm²/h) |
|---|---|---|---|---|---|
| IMC-MP@GCgel | 1.39 ± 0.203 | 0.71 ± 0.104 | 0.51 ± 0.084 | 0.96 ± 0.03 | 0.97 ± 0.02 |
| IMC-NP@GCgel | 3.69 ± 0.239* | 1.88 ± 0.122* | 1.98 ± 0.224* | 1.08 ± 0.15 | 0.67 ± 0.06 |

n = 5. *$P < 0.05$ vs. IMC-MP@GCgel for each category (Student's t-test).

**Table 3. Pharmacokinetic analysis of percutaneous absorption of the IMC@GCgel.**

| Formulation | $ka$ (h⁻¹) | $F$ (×10⁻²) | $AUC_{0-24h\ plasma}$ (nmol·h/mL) |
|---|---|---|---|
| IMC-MP@GCgel | 0.284 ± 0.139 | 1.2 ± 0.21 | 22.6 ± 1.12 |
| IMC-NP@GCgel | 0.353 ± 0.095 | 1.0 ± 0.06 | 20.8 ± 2.26 |

The $k_e$ was 0.05 ± 0.07 h⁻¹. n=5.

the Carbopol® 934 with GA may also be a suitable base material for gelation of IMC NPs. Thus, to assess the crystalline structure before and after bead milling, the XRD was utilized, while the characterization was evaluated using TG-DTA in this study. Although these measurements allow for the assessment of characterization of IMC@GCgel, Fourier Transform Infrared Spectroscopy (FTIR) data are usefulness to provide detail information for the compatibility of the polymer with IMC. Therefore, the FTIR measurements will be necessary in future studies.

Next, we demonstrated the *in vitro* and *in vivo* evaluation of drug release and skin permeability. The drug-release properties of the gel formulations were evaluated using an *in vitro* membrane-based drug-release test. Compared with the IMC-MP@GCgel group, the IMC-NP@GCgel group demonstrated a significantly higher drug release (Fig 4). The membrane pore size used in this study was 450 nm, which is larger than the drug particle size after bead milling. This suggested that the enhanced release from the IMC-NP@GCgel group was due to the passage of NPs through the membrane after being released from the gel, as the particle size of the IMC released from the gel was nanoscale (Fig 4C and 4D). In contrast, the released IMC NCs ranged in size from 40 to 430 nm, larger than their original size in the IMC-NP@GCgel formulation. A Franz diffusion cell containing phosphate buffer was used to assess the release of IMC from the IMC-NP@GC gel. The lower viscosity of the buffer compared with that of the gel contributed to the reduced stability of the IMC NCs, potentially leading to their aggregation and resulting in an increased particle size. Additionally, nanosizing

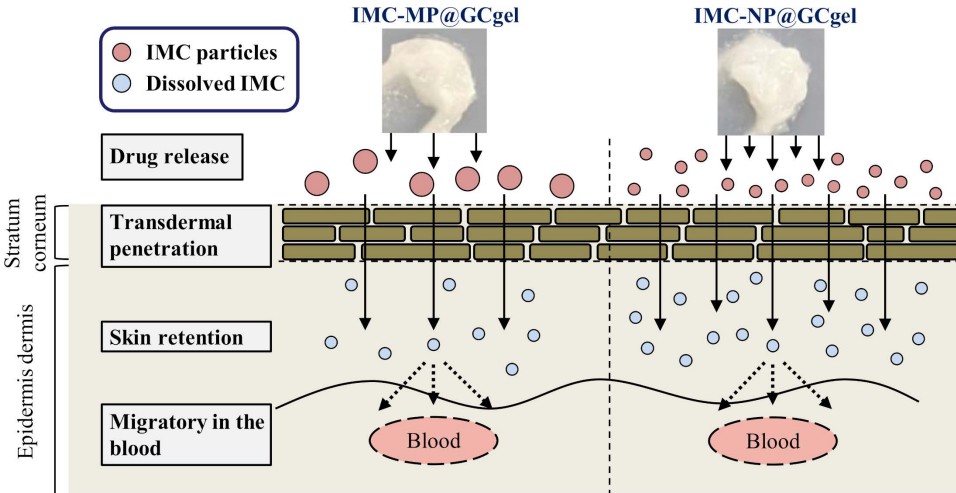

**Fig 7. Scheme for absorption after treatment with IMC-MP@GCgel and IMC-NP@GCgel in the rat skin.**

increases the specific surface area of drug particles, which enhances dissolution rates. The observed increase in the dissolution rate likely contributed to the superior drug release properties of the IMC-NP@GCgel group compared with the IMC-MP@GCgel group.

Moreover, in the *in vitro* skin penetration study, the IMC can pass through the skin, and the $AUC_{0-24h\ penetration}$ of the IMC-NP@GCgel group was 2.57-fold higher than that of the IMC-MP@GCgel group (Fig 5B). Furthermore, the skin permeation parameters ($J_c$, $K_p$, and $K_m$) of the IMC-NP@GCgel group were 2.7-, 2.6-, and 3.9-folds higher than those of the IMC-MP@GCgel group, respectively (Table 2). In addition, skin retention studies further confirmed this trend, showing that the IMC-NP@GCgel group had a significantly higher skin retention than the IMC-MP@GCgel group (Fig 6B). Conversely, *in vivo* evaluation of drug migration into rat plasma revealed no significant differences between the IMC-MP@GCgel and IMC-NP@GCgel groups in terms of drug plasma migration (Fig 6A) or $AUC_{0-24h\ plasma}$ (Table 3). Previously, our group demonstrated that IMC NP-containing gels with MC as an additive (with a particle size distribution comparable to that of IMC used in this study) showed increased drug release, skin permeability, and plasma migration following wet milling [16,17]. However, the current study, which used GA as an additive instead of MC, yielded different results. While skin retention improved, plasma migration did not. Furthermore, the IMC NPs-incorporating gels with GA exhibited higher skin permeation compared to IMC NPs-incorporating gels with MC as the additive ($J_c$ 158 nmol/cm²/h, $K_p$ 5.7 × 10⁻⁴ cm/h, $K_m$ 10.8 × 10⁻², τ 2.3 h, and D 3.7 × 10⁻⁴ cm²/h) (Table 2) [16,17]. This discrepancy likely stems from the difference in the additives (GA versus MC). The uptake of polymeric nanoparticles by cells is influenced by various factors such as particle size, surface characteristics, and the type of cells involved. NCs possess a high surface-to-volume ratio, making their surface properties a key factor in determining their behavior in biological systems [36]. Owing to their elevated surface energies, NCs can interact with, adsorb, or transport different substances, including drugs, proteins, and chemicals, and often exhibit unique catalytic activities [37,38]. Their small dimensions (ranging from approximately 40–500 nm) enable NCs to adhere to SC [39,40]. However, NCs larger than 100 nm cannot easily diffuse through the tissues. In this study, the $J_c$, $K_p$, $K_m$, and D values of the IMC-NP@GCgel were 23.3, 3.3, 1.8, and 1.8, respectively, for the IMC NP-incorporating gels with MC. Therefore, we hypothesized that the addition of GA increases its affinity for skin tissue and IMC, and that the enhanced affinity between GA and the skin may influence the behavior of IMC within the skin pathway. These findings suggest that the choice and changes in additives play a critical role in influencing the drug kinetics and skin permeation properties of the drug NP-incorporating gels. However, further studies are needed to elucidate

this phenomenon for the drug NPs in Carbopol® 934 with GA. Nanoparticle uptake occurs via various pathways, including passive diffusion, receptor-mediated endocytosis, and phagocytosis/micropinocytosis. Numerous studies have demonstrated that endocytosis of nanoparticles is influenced by their size [41–43]. Therefore, in future studies, we aim to demonstrate the relationship between drug absorption and endocytosis.

## Conclusions

In this study, we developed and evaluated a novel gel formulation incorporating IMC NCs to enhance skin permeability and localized drug delivery. The IMC-NP@GC gel with an average particle size of 109.78 nm, was prepared using wet bead milling with GA as an additive. *In vitro* and *in vivo* evaluations revealed superior drug release and skin permeability for the IMC-NP@GC gel formulation compared to its MPs counterpart (IMC-MP@GCgel). Moreover, skin permeability studies showed that the IMC-NP@GCgel formulation achieved a higher dermal absorption than the IMC-MP@GCgel. Skin retention was markedly improved by the NCs formulation. However, drug migration into the plasma was not significantly different between the formulations, suggesting that GA promotes localized drug delivery to IMC NCs rather than systemic absorption (Fig 7). These results provide useful information for the development of effective TDDS based on NCs.

The addition of GA supports the absorption of IMC NPs and enhances the skin retention of the drug without increasing plasma IMC levels.

## Supporting information

**S1 Fig. Minimal data of Fig1D-1.** Full image of IMC-MP@GCgel shown in Fig 1D.
(TIF)

**S2 Fig. Minimal data of Fig1D-2.** Full image of IMC-NP@GCgel shown in Fig 1D.
(TIF)

**S1 Table. Minimal data set of Fig 2A and 2B.** Raw data in Fig 2A and 2B.
(PDF)

**S2 Table. Minimal data set of Fig 2C and 2D.** Raw data in Fig 2C and 2D.
(PDF)

**S3 Table Minimal data set of Fig 1 and 3–6.** Raw data in Figs 1 and 3–6.
(PDF)

## Author contributions

**Conceptualization:** Noriaki Nagai.

**Data curation:** Hiroko Otake, Ogata Fumihiko, Yosuke Nakazawa, Manju Misra.

**Formal analysis:** Hiroko Otake, Ogata Fumihiko, Yosuke Nakazawa, Masanobu Tsubaki.

**Funding acquisition:** Noriaki Nagai.

**Investigation:** Hiroko Otake, Ogata Fumihiko, Yosuke Nakazawa, Manju Misra, Noriaki Nagai.

**Methodology:** Hiroko Otake, Masanobu Tsubaki, Naohito Kawasaki, Noriaki Nagai.

**Project administration:** Naohito Kawasaki, Noriaki Nagai.

**Supervision:** Noriaki Nagai.

**Writing – original draft:** Noriaki Nagai.

**Writing – review & editing:** Noriaki Nagai.

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
