## [Decision Letter · Decision Letter 0]

PONE-D-25-07074Dermal formulation based on carbopol and gum arabic improves skin retention of indomethacinPLOS ONE

Dear Dr. Nagai,

Thank you for submitting your manuscript to PLOS ONE. After careful consideration, we feel that it has merit but does not fully meet PLOS ONE’s publication criteria as it currently stands. Therefore, we invite you to submit a revised version of the manuscript that addresses the points raised during the review process. My personal evaluation of the manuscript, in line with those of the reviewers, is that with a careful and well performed revision of the issues raised the manuscript could be a valuable addition to PLOS One

We look forward to receiving your revised manuscript.

Kind regards,

Giovanni Signore

Academic Editor

PLOS ONE

Journal Requirements:

2**. ** We note that your Data Availability Statement is currently as follows: All relevant data are within the manuscript and its Supporting Information files.

Reviewers' comments:

Reviewer's Responses to Questions

**Comments to the Author**

1. Is the manuscript technically sound, and do the data support the conclusions?

Reviewer #1: No

Reviewer #2: Partly

2. Has the statistical analysis been performed appropriately and rigorously? 

Reviewer #1: Yes

Reviewer #2: No

3. Have the authors made all data underlying the findings in their manuscript fully available?

Reviewer #1: No

Reviewer #2: Yes

4. Is the manuscript presented in an intelligible fashion and written in standard English?

Reviewer #1: No

Reviewer #2: Yes

5. Review Comments to the Author

Reviewer #1: The manuscript attempts to evaluate an indomethacin gel loaded with microparticles and nanoparticles; however, the study lacks scientific rigor and is not well-supported by valid reasoning from previous literature. Several major issues need to be addressed before this work can be considered for publication.

(1) Lack of Justification for Polymer Selection and Comparison

The study primarily focuses on the effect of a polymer, but the justification for its selection is weak. Instead of comparing your results to well-established studies in the literature, you have only compared them with your previous work. This approach does not provide a strong scientific basis for the study.

(2) Poorly Written Results Section

The results are not well-detailed, particularly for thermogravimetric analysis (TGA). There is no in-depth discussion of thermal degradation, weight loss patterns, or any comparative analysis. Such details are essential to support the stability of the formulation.

(3) Lack of Polymer-Drug Compatibility Data

Fourier Transform Infrared Spectroscopy (FTIR) data is missing or inadequately discussed. FTIR is a crucial tool for assessing polymer-drug interactions, and without this analysis, the compatibility of the polymer with indomethacin remains uncertain.

(4) Incomplete Evaluation of Formulation Safety and Stability

Skin irritation studies are absent, which is a significant limitation since any topical formulation must be evaluated for its safety on the skin.

Stability data is also missing. Long-term and short-term stability studies are essential to establish the robustness of the formulation over time.

(5) Image quality

The figure quality is poor and the formulation pictures are blurred. Also, the graphical abstract is not self-explanatory. The different size of arrows will be difficult to elucidate by a young researcher or new reader.

Overall, the manuscript is weak and does not provide a comprehensive or scientifically sound evaluation of the formulation. The lack of literature-based justification, inadequate results, missing polymer-drug compatibility studies, and absence of crucial safety and stability data significantly weaken the study. The manuscript requires substantial revisions before it can be considered for further review.

Reviewer #2: The study could contribute to pharmaceutical sciences and dermatological formulations by providing an improved method for delivering indomethacin through the skin. In this paper, authors evaluated penetration, retention, drug delivery and systemic absorption of indomethacin, using two different gel formulations, based on microparticles or nanoparticles.

The claim is well supported by the current literature.

The details of the methodology are sufficient to allow the experiments to be reproduced.

Experiments are appropriate and well-designed, although data analysis is not always rigorous, leading to conclusions that are not fully substantiated. Although in its current form the paper is unsuitable for publication, it shows good potential and authors are encouraged to resubmit a revised version.

The manuscript is overall well organized and written in a clear scientific English. A certificate of editing is available.

Please, see the attached document for further comments and requests of review.

6. PLOS authors have the option to publish the peer review history of their article (what does this mean? ). If published, this will include your full peer review and any attached files.

**Do you want your identity to be public for this peer review?** For information about this choice, including consent withdrawal, please see our Privacy Policy .

Reviewer #1: No

Reviewer #2: **Yes: ** Gemma Sardelli, PhD

---

## [Author Response · Author response to Decision Letter 1]

4 Apr 2025

We carefully revised our manuscript according to the suggestions of the reviewers, and details are as follows.

<Q and A for Reviewer 1>

Q1. Lack of Justification for Polymer Selection and Comparison: The study primarily focuses on the effect of a polymer, but the justification for its selection is weak. Instead of comparing your results to well-established studies in the literature, you have only compared them with your previous work. This approach does not provide a strong scientific basis for the study.

A1. These are excellent points. This study employs the bead milling method to prepare drug nanocrystals using gum arabic as an additive, and evaluates the dermal drug behavior when these nanocrystals are incorporated into a Carbopol gel. To clarify the research purpose, we have added a statement and reference on the usefulness of Carbopol gel as a base in the development of dermal formulations. These revisions have addressed the previously noted lack of justification for polymer selection and comparison. We appreciate your valuable feedback (line 85-92, Reference 19-22).

Reference

Hatem S, Kamel AO, Elkheshen SA, Nasr M, Moftah NH, Ragai MH, et al. Nano-vesicular systems for melanocytes targeting and melasma treatment: In-vitro characterization, ex-vivo skin retention, and preliminary clinical appraisal. Int J Pharm. 2024;665:124731. doi: 10.1016/j.ijpharm.2024.124731.

Salem HF, Abd El-Maboud MM, Said ASA, Salem MN, Sabry D, Hussain N, et al. Nano Methotrexate versus Methotrexate in Targeting Rheumatoid Arthritis. Pharmaceuticals (Basel). 2022;16(1):60. doi: 10.3390/ph16010060.

Patel V, Mehta TA. Betamethasone Dipropionate Nanocrystals: Investigation, Feasibility and In Vitro Evaluation. AAPS PharmSciTech. 2022;23(6) :197. doi: 10.1208/s12249-022-02346-1.

Khan D, Ahmed N, Muhammad A, Shah KU, Mir M, Rehman AU. A macromolecule infliximab loaded reverse nanomicelles-based transdermal hydrogel: An innovative approach against rheumatoid arthritis. Biomater Adv. 2025;167:214093. doi: 10.1016/j.bioadv.2024.214093.

Q2. Poorly Written Results Section: The results are not well-detailed, particularly for thermogravimetric analysis (TGA). There is no in-depth discussion of thermal degradation, weight loss patterns, or any comparative analysis. Such details are essential to support the stability of the formulation.

A2. In order to respond to the comment, we added the TG data, and discussed for thermal degradation, weight loss patterns, and any comparative analysis in the Result. Thank you for pointing this out (line 340-347, Figure 2 and Figure 2 legend).

Q3. Lack of Polymer-Drug Compatibility Data: Fourier Transform Infrared Spectroscopy (FTIR) data is missing or inadequately discussed. FTIR is a crucial tool for assessing polymer-drug interactions, and without this analysis, the compatibility of the polymer with indomethacin remains uncertain.

A3. The reviewer’s comments are very important. In this study, the bead milling method was employed to prepare drug nanocrystals using gum arabic as an additive, and the transdermal drug behavior of these nanocrystals incorporated into a Carbopol gel was evaluated. To assess the crystalline structure before and after bead milling, X-ray diffraction (XRD) was utilized, while the physical properties were evaluated using differential thermal analysis (TG-DTA). Although these measurements allow for the assessment of physicochemical properties, Fourier Transform Infrared Spectroscopy (FTIR) data are usefulness to evaluate the compatibility of the polymer with indomethacin. In accordance with the given suggestions, we have added the TG data (Figure 2D). Moreover, we added the necessary of FTIR results in future studies. Thank you very much for pointing this out (line 486-491, Figure 2D).

Q4. Incomplete Evaluation of Formulation Safety and Stability: Skin irritation studies are absent, which is a significant limitation since any topical formulation must be evaluated for its safety on the skin.

A4. These are excellent points. In order to respond to the comment, we added the result of skin irritation (line 227-234, 357-358, 483-484, Table 1).

Q5. Stability data is also missing. Long-term and short-term stability studies are essential to establish the robustness of the formulation over time.

A5. In order to respond to the comment, we added the results on the degradation stability and viscosity changes after 1 months of storage in a sealed and dark environment (line 222-225, 354-357, 481-483, Table 1).

Q6. Image quality: The figure quality is poor and the formulation pictures are blurred. Also, the graphical abstract is not self-explanatory. The different size of arrows will be difficult to elucidate by a young researcher or new reader

A6. The reviewer’s comment is correct. We revised the figure quality and graphical abstract (Figure 8). Briefly, the size and shape of the arrows were standardized, and the differences in drug release and permeability were represented by the number of arrows (Figure 8). In addition, we also exchanged the formulation pictures in Figure 1D and Figure 8. Thank you very much for pointing this out (Figure 1D and Figure 8).

Thank you for great comments.

<Q and A for Reviewer 2>

Q1. Line 80: incomplete sentence. Please, revise for clarity.

A1. The reviewer’s comment is correct. We revised this sentence (line 80-81).

Q2. Lines 318-319: The claim that particles in IMP-NC are more evenly distributed than in IMP-MC is not sufficiently supported by Image 1D alone. Further justification or quantitative analysis is needed.

A2. The reviewer’s comments are very important. We have replaced the formulation images with higher-resolution versions (Figure 1D). In addition, the quantitative analysis of drug uniformity in the gel were shown in Figure 3A. We added this information in the relevant part (line 337-339, Figure 1D and 3A).

Q3. Line 335: a scale bar should be added to figure 1D.

A3. In order to respond to the comment, we added the scale bar (5 mm) in the Figure 1D (Figure 1D).

Q4. Line 372: Instead of a multiple comparison, compare each parameter of the two formulations individually using a Student’s t-test or a non-parametric test if normality is not satisfied.

A4. The reviewer’s comment is correct. We compared each parameter of the two formulations individually by using a Student’s t-test (line 413).

Q5. Line 376: The text (lines 365-366) states that no significant differences were observed in the reported parameters between the two formulations, but the statistical test you used is not specified. Please, clarify.

A5. In order to respond to the comment, we compared each parameter of the two formulations individually by using a Student’s t-test. We mentioned the information in the text (line 407).

Q6. Line 385: ANOVA One-Way with Tukey-Kramer post hoc test is not suitable in this context, since measurements overtime are not independent (aliquots at different time points come from the same sample, thus they are dependent). A repeated-measures ANOVA should be used if normality is satisfied.

A6. The reviewer’s comment is correct. We reanalyzed the statistical analysis by using JMP software version 5.1 (SAS Institute). In order to respond to the comment, we compared each parameter of the two formulations individually by using a repeated-measures ANOVA. We mentioned the information in the text (line 322-324, 426).

Q7. Line 391: The caption should specify that this is the in vitro approach.

A7. These are excellent points. We have explicitly stated in the caption that this is an in vitro approach (line 431, 434).

Q8. Line 393: As above, ANOVA One-Way with Tukey-Kramer post hoc test is inappropriate, since also these measurements are not independent each other. Use repeated-measures ANOVA if normality is met.

A8. The reviewer’s comments are very important. In order to respond to the comment, we evaluated the statistical analysis using a repeated-measures ANOVA, and mentioned the information in the text (line 322-324, 434-435).

Q9. Lines 395-397: You would put this sentence in the main text and discuss its implication, as it suggests that that the IMC (but not the carrier) can pass through the skin, at least in vitro.

A9. In accordance with the suggestions, we mentioned that, the IMC can pass through the skin, at least in vitro. Thank you for pointing this out (line 399, 402-404, 508).

Q10. Line 401: The caption is inconsistent with the graph (Fig. 6A), which shows a time scale rather than plasma IMC levels 6 hours after treatment. Please correct this inconsistency.

A10. The reviewer’s comment is correct. We corrected to “24 h” from “6 h” (line 442).

Thank you for great comments.

---

## [Decision Letter · Decision Letter 1]

PONE-D-25-07074R1Dermal formulation based on carbopol and gum arabic improves skin retention of indomethacinPLOS ONE

Dear Dr. Nagai,

Thank you for submitting your manuscript to PLOS ONE. After careful consideration, we feel that it has merit but does not fully meet PLOS ONE’s publication criteria as it currently stands. Therefore, we invite you to submit a revised version of the manuscript that addresses the points raised during the review process.i agree with the reviewer that a further check should be performed on the result of the statistical analysis provided, to rule out the possibility for mistakes.

We look forward to receiving your revised manuscript.

Kind regards,

Giovanni Signore

Academic Editor

PLOS ONE

Reviewers' comments:

Reviewer's Responses to Questions

**Comments to the Author**

1. If the authors have adequately addressed your comments raised in a previous round of review and you feel that this manuscript is now acceptable for publication, you may indicate that here to bypass the “Comments to the Author” section, enter your conflict of interest statement in the “Confidential to Editor” section, and submit your "Accept" recommendation.

Reviewer #2: (No Response)

2. Is the manuscript technically sound, and do the data support the conclusions?

Reviewer #2: Yes

3. Has the statistical analysis been performed appropriately and rigorously? 

Reviewer #2: I Don't Know

4. Have the authors made all data underlying the findings in their manuscript fully available?

Reviewer #2: Yes

5. Is the manuscript presented in an intelligible fashion and written in standard English?

Reviewer #2: Yes

6. Review Comments to the Author

Reviewer #2: The authors have addressed all the comments raised in the first round of review. Although the request to adopt a different and more appropriate method of data analysis has been accepted and the changes have been implemented in the main text and/or figure caption, the statistical significance remains exactly the same as in the previous, incorrect analysis. The authors are kindly requested to check for any typos or inconsistencies in the revised version of the manuscript, or to demonstrate the validity of the results obtained from the newly performed analysis.

7. PLOS authors have the option to publish the peer review history of their article (what does this mean? ). If published, this will include your full peer review and any attached files.

**Do you want your identity to be public for this peer review?** For information about this choice, including consent withdrawal, please see our Privacy Policy .

Reviewer #2: **Yes: ** Gemma Sardelli

---

## [Author Response · Author response to Decision Letter 2]

1 May 2025

We carefully revised our manuscript according to the suggestions of the reviewer 2, and details are as follows.

<Q and A for Reviewer 2>

Q1. The authors have addressed all the comments raised in the first round of review. Although the request to adopt a different and more appropriate method of data analysis has been accepted and the changes have been implemented in the main text and/or figure caption, the statistical significance remains exactly the same as in the previous, incorrect analysis. The authors are kindly requested to check for any typos or inconsistencies in the revised version of the manuscript, or to demonstrate the validity of the results obtained from the newly performed analysis.

A1. The reviewer’s comment is correct. We have appropriately corrected the symbols in Figures 4A and 5A. In addition, we have revised the corresponding descriptions in the manuscript, and carefully checked that the content is accurately reflected. Thank you very much for pointing this out (line 322, 424, 432, Figure 4A and 5A).

Thank you for great comments.

---

## [Decision Letter · Decision Letter 2]

Dermal formulation based on carbopol and gum arabic improves skin retention of indomethacin

PONE-D-25-07074R2

Dear Dr. Nagai,

We’re pleased to inform you that your manuscript has been judged scientifically suitable for publication and will be formally accepted for publication once it meets all outstanding technical requirements.

Kind regards,

Giovanni Signore

Academic Editor

PLOS ONE

Additional Editor Comments (optional):

Reviewers' comments:

Reviewer's Responses to Questions

**Comments to the Author**

1. If the authors have adequately addressed your comments raised in a previous round of review and you feel that this manuscript is now acceptable for publication, you may indicate that here to bypass the “Comments to the Author” section, enter your conflict of interest statement in the “Confidential to Editor” section, and submit your "Accept" recommendation.

Reviewer #2: All comments have been addressed

2. Is the manuscript technically sound, and do the data support the conclusions?

Reviewer #2: (No Response)

3. Has the statistical analysis been performed appropriately and rigorously? 

Reviewer #2: (No Response)

4. Have the authors made all data underlying the findings in their manuscript fully available?

Reviewer #2: (No Response)

5. Is the manuscript presented in an intelligible fashion and written in standard English?

Reviewer #2: (No Response)

6. Review Comments to the Author

Reviewer #2: The authors have met all the requirements, and the graphs have been appropriately adjusted in line with the revised statistical analysis.

7. PLOS authors have the option to publish the peer review history of their article (what does this mean? ). If published, this will include your full peer review and any attached files.

**Do you want your identity to be public for this peer review?** For information about this choice, including consent withdrawal, please see our Privacy Policy .

Reviewer #2: **Yes: ** Gemma Sardelli

---

## [Editor Report · Acceptance letter]

PONE-D-25-07074R2

PLOS ONE

Dear Dr. Nagai,

I'm pleased to inform you that your manuscript has been deemed suitable for publication in PLOS ONE. Congratulations! Your manuscript is now being handed over to our production team.

Kind regards,

on behalf of

Dr. Giovanni Signore

Academic Editor

PLOS ONE